# Compendium-Wide Analysis of *Pseudomonas aeruginosa* Core and Accessory Genes Reveals Transcriptional Patterns across Strains PAO1 and PA14

Alexandra J. Lee,[a] Georgia Doing,[b,c] Samuel L. Neff,[b] Taylor Reiter,[e] Deborah A. Hogan,[b] Casey S. Greene[d,e]

[a]Genomics and Computational Biology Graduate Program, University of Pennsylvania, Philadelphia, Pennsylvania, USA
[b]Department of Microbiology and Immunology, Geisel School of Medicine at Dartmouth, Hanover, New Hampshire, USA
[c]The Jackson Laboratory for Genomic Medicine, Farmington, Connecticut, USA
[d]Department of Pharmacology, University of Colorado School of Medicine, Denver, Colorado, USA
[e]Department of Biochemistry and Molecular Genetics, University of Colorado School of Medicine, Denver, Colorado, USA

**ABSTRACT** *Pseudomonas aeruginosa* is an opportunistic pathogen that causes difficult-to-treat infections. Two well-studied divergent *P. aeruginosa* strain types, PAO1 and PA14, have significant genomic heterogeneity, including diverse accessory genes present in only some strains. Genome content comparisons find core genes that are conserved across both PAO1 and PA14 strains and accessory genes that are present in only a subset of PAO1 and PA14 strains. Here, we use recently assembled transcriptome compendia of publicly available *P. aeruginosa* RNA sequencing (RNA-seq) samples to create two smaller compendia consisting of only strain PAO1 or strain PA14 samples with each aligned to their cognate reference genome. We confirmed strain annotations and identified other samples for inclusion by assessing each sample's median expression of PAO1-only or PA14-only accessory genes. We then compared the patterns of core gene expression in each strain. To do so, we developed a method by which we analyzed genes in terms of which genes showed similar expression patterns across strain types. We found that some core genes had consistent correlated expression patterns across both compendia, while others were less stable in an interstrain comparison. For each accessory gene, we also determined core genes with correlated expression patterns. We found that stable core genes had fewer coexpressed neighbors that were accessory genes. Overall, this approach for analyzing expression patterns across strain types can be extended to other groups of genes, like phage genes, or applied for analyzing patterns beyond groups of strains, such as samples with different traits, to reveal a deeper understanding of regulation.

**IMPORTANCE** *Pseudomonas aeruginosa* is a ubiquitous pathogen. There is much diversity among *P. aeruginosa* strains, including two divergent but well-studied strains, PAO1 and PA14. Understanding how these different strain-level traits manifest is important for identifying targets that regulate different traits of interest. With the availability of thousands of PAO1 and PA14 samples, we created two strain-specific RNA-seq compendia where each one contains hundreds of samples from PAO1 or PA14 strains and used them to compare the expression patterns of core genes that are conserved in both strain types and to determine which core genes have expression patterns that are similar to those of accessory genes that are unique to one strain or the other using an approach that we developed. We found a subset of core genes with different transcriptional patterns across PAO1 and PA14 strains and identified those core genes with expression patterns similar to those of strain-specific accessory genes.

**KEYWORDS** *Pseudomonas aeruginosa*, compendia, transcriptome

Address correspondence to Casey S. Greene, casey.s.greene@cuanschutz.edu.

The authors declare no conflict of interest.

For a companion article on this topic, see https://doi.org/10.1128/mSystems.00341-22.

*P*seudomonas aeruginosa is a Gram-negative bacterium that is able to thrive in a variety of different abiotic environments, including soil and water, and live in association with plants and animals (1). *P. aeruginosa* is also an opportunistic human pathogen that is frequently implicated in hospital-acquired infections (2, 3) and is a particular concern for immunosuppressed and vulnerable individuals (4, 5).

Among *P. aeruginosa* strains, there is much phenotypic diversity. Clinical and environmental strains exhibit various capacities for biofilm formation (6, 7), levels of antibiotic susceptibility (8, 9), metabolic profiles (10), and differences in virulence factor production (11). Some strains can also perform biotransformations of xenobiotic compounds (12, 13). This strain-level phenotypic diversity is reflected in its genetic diversity; the *P. aeruginosa* genome contains both conserved core genes and strain-specific accessory genes (14, 15). A phylogenetic analysis across 1,311 strains, using only core genes, divided *P. aeruginosa* strains into five major lineages (14). Within these five lineages, there were two predominant lineages that contain one each of the commonly used laboratory strains PAO1 and PA14 (16). Compared to strain PAO1, strain PA14 was found to be more virulent in a number of model systems (17, 18), and comparative genomic studies between PAO1 and PA14 found that the differences in virulence were combinatorial (18–20) and only partially attributable to the presence or expression of strain-specific accessory genes (17, 19, 21–23). Core genes can drive differences in virulence across strains if there is strain-specific regulation. For example, while both PAO1 and PA14 contain the same molecular machinery for surface sensing and attachment, they activate distinct surface-sensing signaling circuits that contribute to differences in cell surface attachment (24–26). These data highlight how the same factors can be deployed in different ways.

Many transcription factors (TFs) and other transcriptional regulatory elements found in *P. aeruginosa* control the expression of many gene products that mediate different traits (27, 28) such as virulence (29–31), and these can vary across strains. Strain-level differences in expression between PAO1 and PA14 have recently been directly interrogated under a single condition by Sana et al., who found a set of core genes that were differentially expressed in PAO1 compared to PA14 (32). Many of these genes were involved in quorum sensing (QS), which is a cell-cell signaling communication system that regulates many virulence factors (33). This difference is consistent with differences in QS induction kinetics that have been observed previously (34). This study demonstrates the need for further investigation into strain-based differences in the expression of both core and accessory genes to understand how they work together in orchestrating virulence. Another example is within the type III secretion system (T3SS), which is a virulence determinant that allows *P. aeruginosa* to deliver toxic effector proteins to host cells. Strains PAO1 and PA14 both secrete the effectors ExoT and ExoY, but they also differ in that PAO1 secretes ExoS and PA14 secretes ExoU (35). The production of these accessory virulence factors was found to be regulated by strain-specific response regulators (36) as well as shared regulators like ExsA (37). The secretion of ExoU was found to increase lethality in mice more than the secretion of ExoS (38). Overall, ExoS and ExoU affect different host pathways and illustrate important consequences of differences in accessory gene expression driven by the strain-specific regulation of both core and accessory genes (39).

In general, transcriptional regulatory networks are known to be versatile across *P. aeruginosa* strains, particularly PAO1, PA14, and PAK (36). This versatility was observed across strains within a single species in other microbes as well (40–42). These network differences between strains were due, in part, to the accessory genome: there were some strain-specific TFs, target genes (i.e., *exoS* and *exoU*), and TF-target gene interactions (36, 40, 41). Overall, the above-mentioned examples demonstrated that both core and accessory gene regulation affects strain-level traits like virulence. While the existence of core and accessory genomes has been known, how different genomic backgrounds affect transcriptional profiles remains to be explored. Extending the findings of Trouillon et al., who found that there are some conserved and some different regulatory interactions between PAO1 and PA14 (36), we can examine the downstream

transcriptional patterns across PAO1 and PA14, accounting for the different gene groups.

In this paper, we complement gene content comparisons of strain PAO1 to strain PA14 (17–22, 24–26) by examining the transcriptome differences between strains using compendia of gene expression data. Many comparative studies focus on only a single condition when studying core and accessory gene transcription (22, 43–46); here, we demonstrate that additional information about the differences in transcriptional patterns in strains PAO1 and PA14 for core and accessory genes can be elucidated by comparing expression patterns across many experiments performed by different laboratories (47–51). Leveraging the newly formed *P. aeruginosa* RNA sequencing (RNA-seq) compendia created by Doing et al. in a companion article (52), in which 2,333 RNA-seq samples were aligned to both PAO1 and PA14 cDNA reference genomes, we created two new data sets containing data for PAO1- or PA14-only samples. Using these compendia, we found that among core genes, there is a subset of genes that exhibit stable correlation patterns across strain types based on their transcriptional profiles, while others differ substantially. The most stable core genes are less often coexpressed with accessory genes than the least stable core genes. By enabling a more nuanced understanding of transcriptional behaviors across *P. aeruginosa* strains, starting with PAO1 and PA14 strains, the RNA-seq compendium helps to further elucidate differences in how genes in *P. aeruginosa* are regulated across strain types.

## RESULTS

**Creation of PAO1-specific and PA14-specific compendia that contain samples from diverse experiments.** In the companion article by Doing et al. (52), 2,333 recently downloaded, filtered, and normalized public *P. aeruginosa* RNA-seq samples were separately mapped to a PAO1 cDNA reference genome and a PA14 cDNA reference genome, regardless of the strain annotation, and validated. Because not all samples were annotated with a strain, we predicted the strain used in a given sample based on the expression of accessory genes that were in either PAO1 or PA14 but not both. Genes present in one strain but not the other were defined by the BACTOME database (53). We calculated the median expression level for each sample (Fig. 1A). For all 2,333 samples, the median expression level of PAO1 accessory genes was calculated using the PAO1-mapped RNA-seq compendium, and the median PA14-specific accessory gene expression values were calculated using the PA14-mapped RNA-seq compendium. When all 2,333 samples were plotted based on the median expression levels of PAO1- and PA14-specific accessory genes, it was clear that most samples had high expression levels of only one of the two strain-specific sets. Strain annotations in Sequence Read Archive (SRA) entries were present for approximately 70% of the 2,333 samples, and these annotations strongly supported the use of accessory gene expression as a predictor of strain identity (Fig. 1A) (99% of PAO1 and 98% of PA14 SRA-labeled samples were assigned as PAO1 or PA14 based on expression). A threshold was applied to the median expression level of each set of strain-specific accessory genes, as described in the methods in Data Set S1 at https://osf.io/vdp9u. This threshold was determined based on the value that best separated the distribution of median accessory gene expression values in the 2,333 samples annotated as either PAO1 or PA14 from the distribution in samples annotated as non-PAO1 or non-PA14 (Fig. 1B and C, respectively). This approach allowed us to identify whether unannotated samples were obtained from strains that are either PAO1 (or PAO1-like) or PA14 (or PA14-like) or distinct from both (e.g., clinical isolates). Using these cutoffs, a strain PAO1 sample compendium with 890 samples and a strain PA14 sample compendium with 505 samples were created (see Fig. S1A and B in the supplemental material). It is possible that some clinical isolates with gene profiles very similar to those of either PAO1 or PA14 were included.

Both the strain PAO1 sample compendium and the strain PA14 sample compendium contained samples generated under diverse conditions (Fig. 1F), although LB medium was the predominant medium in both compendia, used for 51% of the samples in the

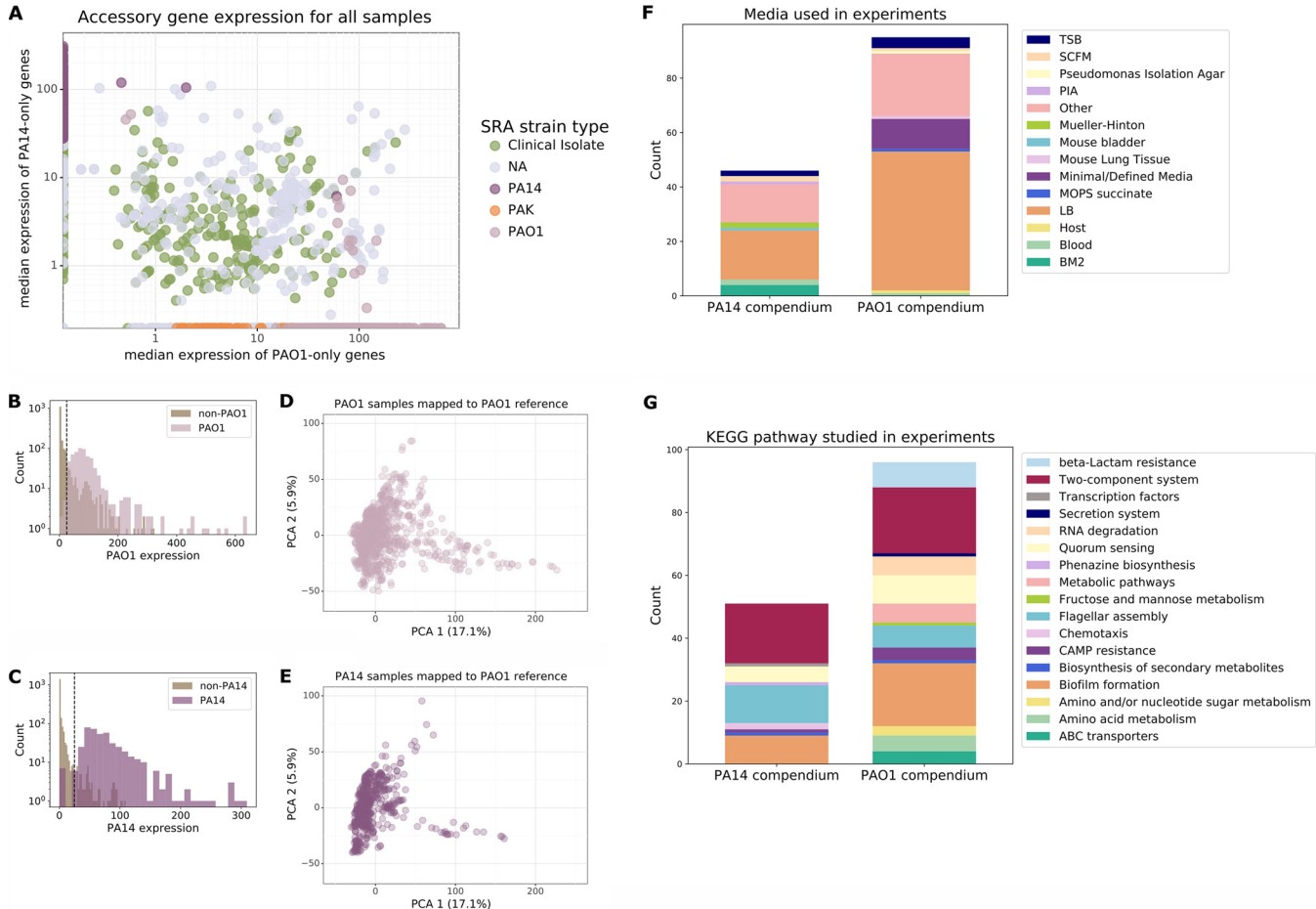

**FIG 1** Creation and composition of PAO1 and PA14 sample compendia. (A) Median accessory gene expression of 2,333 *P. aeruginosa* RNA-seq samples mapped to the PAO1 and PA14 reference genomes, as described in the companion paper by Doing et al. NA, all other strain types (52). (B) Distribution of accessory gene expression in PAO1 SRA-annotated samples versus non-PAO1 SRA-annotated samples. (C) Distribution of accessory gene expression in PA14 SRA-annotated samples versus non-PA14 SRA-annotated samples. (D) PCA of PAO1 sample compendium aligned to the PAO1 reference. (E) The same structure as that in panel D is seen using the PA14 sample compendium aligned to the PAO1 reference. (F) Media or groups of media used in experiments across the two PAO1 (129 experiments with 890 samples) and PA14 (59 experiments with 505 samples) sample compendia. TSB, tryptic soy broth; MOPS, 3-(*N*-morpholino)propanesulfonic acid. BM2, Basal medium with glutamic acid; PIA, Pseudomonas isolation agar. (G) KEGG pathway(s) associated with genes perturbed (i.e., genes "knocked out"/"knocked down" or overexpressed) in experiments, compared across the two compendia. There are some experiments that did not perturb any gene, and there are also some genes that did not have a KEGG annotation; these are not displayed.

strain PAO1 compendium and 39% of the samples in the strain PA14 compendium (Fig. 1F). Many experiments relied on genetic manipulations (e.g., gene deletions or gene overexpression) that targeted genes related to multiple different functions and pathways (Fig. 1G). Interestingly, the KEGG pathway annotations for manipulated genes showed that the three largest categories for both the samples in the strain PAO1 sample compendium and the samples in the strain PA14 sample compendium were biofilm, quorum sensing, and two-component systems, suggesting high-level commonalities among the functions of genes that researchers have interrogated in each strain.

**Comparison of condition-driven differences to strain-driven differences.** Despite the genomic differences between PAO1 and PA14 strains, the differences in the transcriptomes within each strain across many contexts (e.g., growth conditions, genetic manipulations, or other mutations) are much greater than the differences between strains (Fig. 1D and E). The centroid difference between the PAO1 sample compendium and the PA14 sample compendium mapped to the PAO1 reference (including PAO1-specific accessory genes) is 25, much smaller than the spread of the samples within each strain-specific compendium (variance of 32 for PAO1 samples and variance of 27 for PA14 transcriptomes using the PAO1 reference for read mapping). This diversity within strain type is also seen using samples mapped to the PA14 reference (Fig. S1C

and D). Additionally, Doing et al. found that differential expression analysis results were consistent using either the PAO1 or the PA14 reference (52). This diversity among samples within each strain type indicated the value of using these types of gene expression compendia to identify similarities and differences in transcriptional patterns between PA14 and PAO1 across the many contexts that have been studied.

**Certain core genes and pathways are transcriptionally stable across strain types.** Because the strain PAO1 sample compendium and the strain PA14 sample compendium were created using data obtained from different reference genomes, we could not directly correlate gene expression profiles. Thus, we assessed a second-order correlation to determine how similar or different a gene's expression profile is between the two strains. Since core genes are shared by both strain types, we first asked, Do core genes have similar correlation profiles between the two strains? To do so, we used transcriptional stability, defined as the similarity of transcriptional neighbors between homologous core genes in the two compendia; i.e., two homologous core genes are similar if their transcription patterns are most closely correlated with the same set of genes across samples. We first linked PAO1 and PA14 homologs for all 5,349 core genes. For each core gene, we then examined the correlation of their correlations with all other core genes across all samples in each strain-specific compendium. If two homologous core genes had a high second-order correlation, we considered those core genes to have high transcriptional stability (Fig. 2B). After performing this pairwise comparison across all core genes, we defined a subset of genes that were the most stable and another subset of genes that were the least stable by taking the top and bottom 5% of genes based on their transcriptional stability (Fig. 2B; Table S1).

The most stable core genes were significantly enriched in KEGG pathways that represented essential functions: translation, RNA metabolism, DNA repair and recombination, central carbon metabolism (*sdhABCD*, *sucABCD*, and *lpd*) (54), amino acid metabolism, and peptidoglycan biosynthesis (Table S2). These pathways were consistent with the functions of core essential genes identified previously by Poulsen et al. (40% of our most stable core genes were also found to be essential) (55). In addition to these essential pathways, stable core genes also included genes within the type VI secretion system (T6SS) (56), specifically genes in Hcp secretion island I (H1-T6SS). However, many of these T6SS genes had identical paralogs and were thus removed from further discussion (see Data Set S1 in reference 52 for a list of genes with identical or nearly identical paralogs). Since cooperonic genes are cotranscribed, and we might expect that they are "neighbors" (genes with whom they are most highly correlated), we determined if the genes that were the most stable were those in large operons. To determine if stability correlated with membership in an operon, we examined the correlation between the transcriptional stability of a gene and the size of the operon that the gene is contained in. We found that some highly stable genes are not in operons, and there was not a strong correlation between stability and operon size ($R^2 =$ 0.086 and $R^2 = 0.11$ for PAO1 and PA14, respectively) (Fig. S2A and B).

The KEGG pathways significantly enriched in the least stable core genes were genes involved in the biosynthesis of paerucumarin (encoded by the *pvc* genes), which was interesting in light of the result that the expression of the *pvc* gene regulator led to different product ratios in strain PA14 and strain PAO1 (Table S3) (57). In addition, genes related to the degradation of aromatic compounds, including tyrosine and benzoate, were also enriched in the genes that were less stable across strains (Table S4). Given the existence of the least stable core genes, which are inconsistent in their coexpressed genes between strains PAO1 and PA14 (Fig. 2B), we hypothesize that differences in core gene expression contribute to the differences in phenotypes across conditions. For example, differences in pyocyanin and 4-hydroxy-2-heptylquinoline production between PAO1 and PA14 (58, 59) may be due to differences in the levels of aromatic amino acid catabolites that can influence the production of QS molecules and phenazines (60, 61).

As a validation, we recalculated the most and least stable genes using the *P. aeruginosa* microarray gene expression compendium described previously by Tan et al. (47) (Fig. S2C and Table S4). We found a significant ($P = 3.8e-48$) overrepresentation of the "most stable" core genes using the array compendium within the most stable core

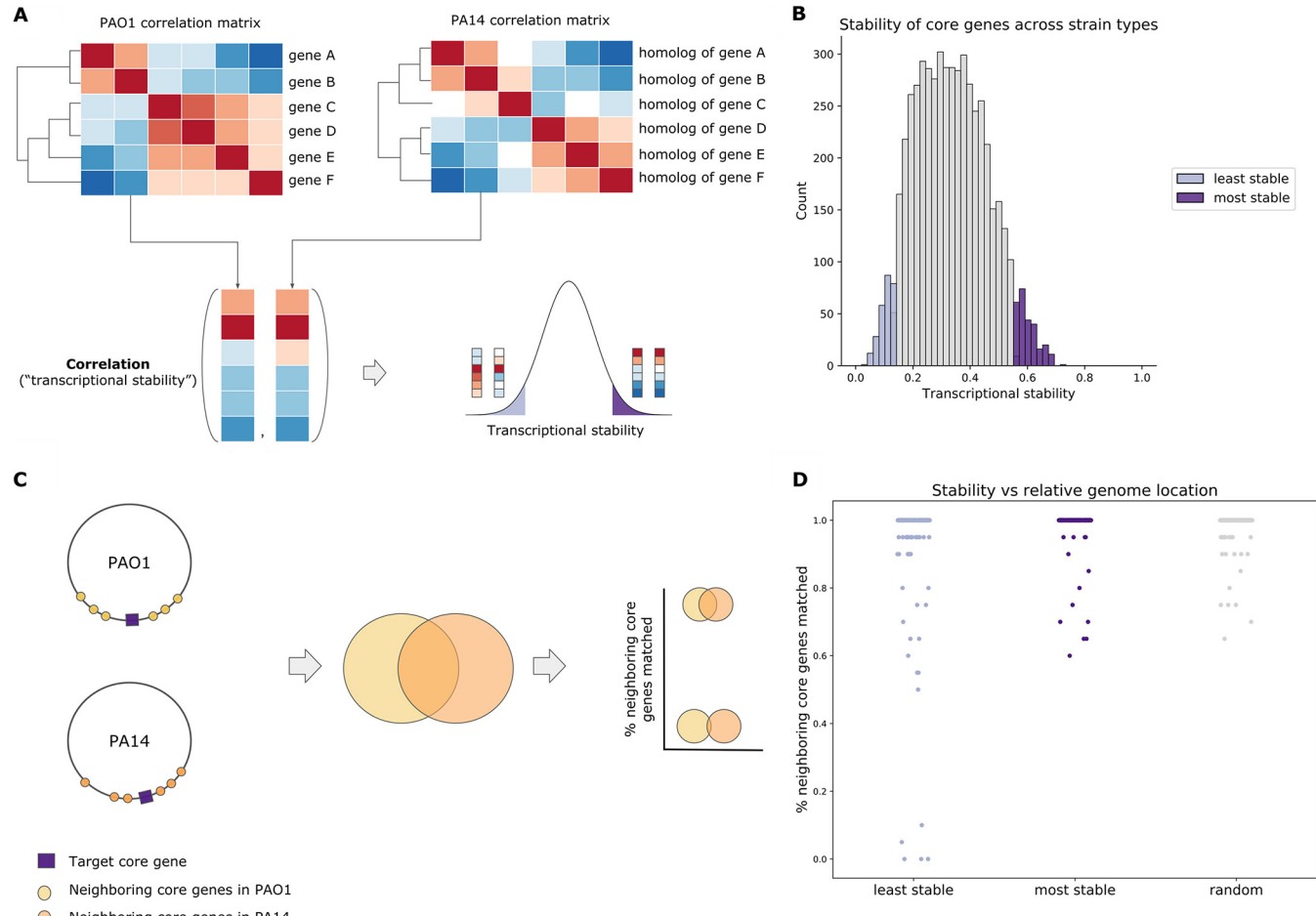

**FIG 2** Relationships between core genes. (A) Workflow describing how the most and least stable core genes were identified by comparing the correlation profiles of homologous gene pairs. (B) Distribution of correlation scores representing how stable core genes are across strains, with the most stable core genes highlighted in dark purple and the least stable core genes in light purple. (C) Workflow to test if the least stable genes are located in different regions of the genome across strains. Given a target least stable or most stable core gene, we compare the overlap of neighbors in PAO1 versus the neighbors in PA14. If the overlap (percentage of neighboring core genes matched) is high, meaning that many of the neighbors are the same in PAO1 and PA14, this would indicate that the gene is located in the same neighborhood in both strains. (D) Consistency of the locations of homologous genes in PAO1 versus PA14 strains among the least stable core genes (light purple), the most stable core genes (dark purple), and a random set of core genes (gray).

genes found using the RNA-seq compendium. Again, the *pvc* genes were among the least stable (Table S3). Overall, we found consistency in the transcriptional stability scores using both the RNA-seq and array compendia ($R^2 = 0.5$) (Fig. S2D). It was also interesting to note that the gene-gene correlation scores within the microarray compendium were lower than those for genes in the RNA-seq compendium.

As genomic rearrangements can place genes in different genomic neighborhoods with different regulatory control, we compared the genomic locations of homologous genes. We determined the percent overlap of neighboring core genes in PAO1 versus PA14 by calculating the percentage of the 10 nearest neighbors in PAO1 that were homologous to the 10 nearest PA14 neighbors. In most cases (262 out of 267 genes), less stable core genes were located in similar locations in PAO1 and PA14 (Fig. 2C and D). Differences in genomic context were observed for PA0982, PA2520 (*czcA*), PA3867, PA2226 (*qsrO*), and PA3507 (52). Overall, this analysis suggests that a change in the genome location was not the primary driver of instability in these least stable genes.

**Core genes that are the most stable are less often coexpressed with accessory genes.** We wanted to determine if differences in core gene stability were associated with coexpression with accessory genes that can be coopted into existing core gene sets and can influence core gene expression (62–64). We determined if transcriptional

stability was associated with coexpression with accessory genes using the core genes in the most and least stable sets. For each gene, we identified the 10 most coexpressed genes for both PAO1 and PA14 and then calculated the proportion of these coexpressed genes that were accessory genes compared to the proportion of accessory genes in the genome (Fig. 3A). We found that the most stable core genes on average exhibit less coexpression with accessory genes: the most stable core genes have fewer transcriptional neighbors that are accessory genes (Fig. 3B).

**Comparison of *exoU* and *exoS* coregulated genes.** We also examined transcriptional stability in the context of the type III secretion system (T3SS), which is a virulence mechanism that allows *P. aeruginosa* to inject toxic effector proteins into the cytoplasm of eukaryotic cells. We selected the T3SS because while there are four effector proteins secreted by this system (ExoU, ExoS, ExoY, and ExoT), the genes encoding ExoU and ExoS are strain specific (i.e., the presence of *exoU* and *exoS* tends to be mutually exclusive in PAO1 and PA14 strains), and they have differential effects on the host (38, 65). Additionally, these two strain-specific effector genes are not syntenic: the two genes located at different genomic locations in their respective strains (*exoU* at PA14_51530, positions 4580957 to 4583020, minus strand, and *exoS* at PA3841, positions 4303141 to 4304502, minus strand) and their neighbors are different. We found 34 core genes that were highly coexpressed with both *exoU* and *exoS*. Some of these were related to the T3SS secretion machinery (*popNBD*, *pcrGDVH*, and *pscCN*) and genes encoding the effector protein ExoT (Fig. 3C). In addition to core genes that were highly coexpressed with both *exoU* and *exoS*, we found some core genes that were highly expressed with only *exoU* or only *exoS*: 22 core genes were highly coexpressed with *exoU*, while 35 core genes were highly coexpressed with *exoS*. Core genes more highly coexpressed with *exoS* were related to metabolic pathways, including *acoRB*, which is involved in acetoin catabolism; *adh*, which is an alcohol dehydrogenase (Table S5); and *spcS*, which encodes an ExoS chaperone. In contrast, genes that were more highly coexpressed with *exoU* included *oprD* (66), which encodes an outer membrane porin implicated in carbapenem resistance, a trait associated with ExoU-positive strains (67). Other genes more coexpressed with *exoU* are *rhlR*, which encodes a regulator involved in quorum sensing in response to a signal generated by the coexpressed synthase RhlI, and *pheC*, which encodes a periplasmic cyclohexadienyl dehydratase involved in phenylalanine biosynthesis. Interestingly, *pheC* is adjacent to the *rhlR-rhlI* genes. Despite ExoS and ExoU being secreted by the same secretion pathway, we identified distinct groups of core genes that might suggest different environments in which the two clades are found: acetoin, a major product of the *Enterobacteriaceae*, may be present when the T3SS is activated for many ExoS-containing strains. These different transcriptional relationships are consistent with the genes having different pathogenic effects on host cell function and being located at different relative locations on the genome (39). Overall, most stable core genes tend to be less coexpressed with accessory genes, suggesting that the interrelationship between core and accessory gene expression depends on the strain type. Therefore, including both core and accessory genes will be important for revealing possible new regulatory relationships in different strain types.

**Clustering by coexpression patterns identified accessory-accessory modules.** We used a graph-based clustering approach, called affinity propagation (68), to cluster accessory genes into modules based on their correlation profiles. In the PAO1 sample compendium, the 202 accessory genes were clustered into 28 modules (Table S6), with clusters ranging from 2 to 53 genes and having a median size of 5 genes. Similarly, the 530 PA14 accessory genes were clustered into 70 modules, with clusters ranging from 3 to 44 genes and having a median size of 7 genes (Table S7). Based on manual inspection, we found that accessory modules tended to correlate with known operons, as expected. The predicted operon for each gene is listed in Tables S6 and S7, and manual inspection highlights that this method correctly identified the coregulation of cooperonic genes. This finding was expected as we confirmed that cooperonic genes were highly correlated in expression with each other in these compendia (52), and in most cases, operon structure is conserved across strains. Indeed, some accessory modules contained only cooperonic genes (e.g., module 13 [*wbp* genes involved in O-antigen

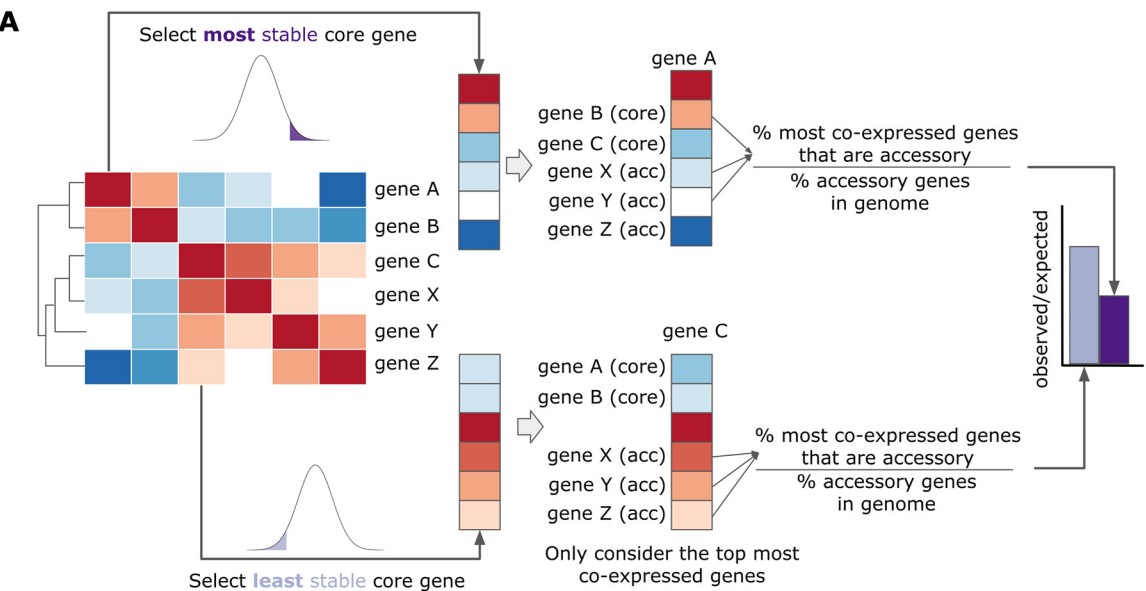

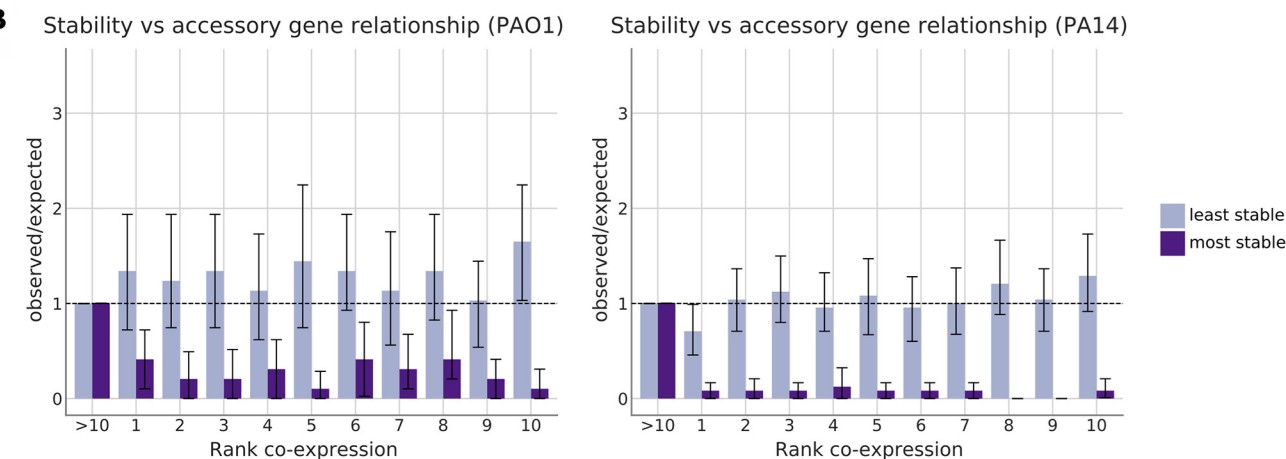

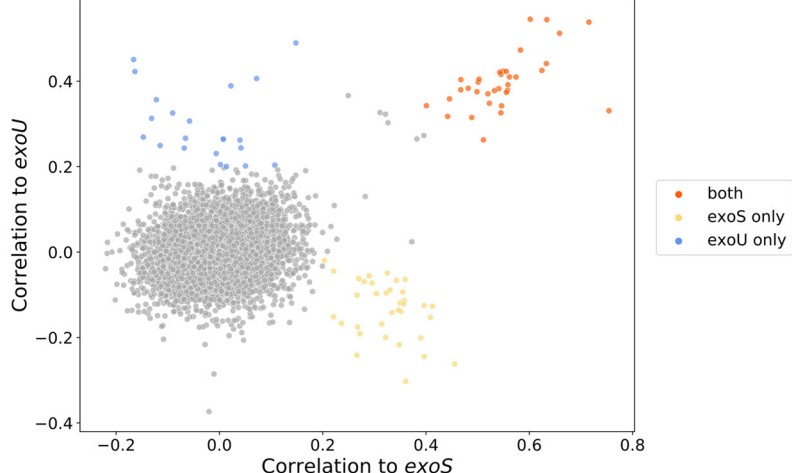

**FIG 3** Relationships between core and accessory genes. (A) Workflow describing how genes coexpressed with the most and least stable core genes were identified. Given a most stable core gene, the top most coexpressed genes were identified, and the proportion of coexpressed genes that are accessory (acc) compared to the expected proportion of accessory genes in the genome was calculated. (B) Proportion of accessory genes that the most stable (dark purple) and least stable (light purple) core genes were coexpressed with. (C) Scatterplot depicting the correlation between core genes and two T3SS accessory genes, *exoS* and *exoU*.

biosynthesis] and module 23 [*psl* genes involved in extracellular matrix production]) (Table S6). However, most modules contain genes from more than one operon. There were multiple modules that contained an uncharacterized or partially characterized transcriptional regulator with noncooperonic genes, which will provide the basis for future mechanistic exploration, such as modules 10, 19 (which contains *vqsM*), 22, and 24 in strain PAO1 (Table S6) and modules 6, 24, and 28 in strain PA14 (Table S7). We also observed that some accessory modules had similar functions in PAO1 and PA14, such as those related to fimbrial biosynthesis and lipopolysaccharide (LPS) biosynthesis. It is worth noting that module 45 for strain PA14 was enriched in genes that may be problematically mapped due to the presence of high-identity paralogs (52), and caution should be used in considering this module as it may reflect a technical feature of the data rather than a biological one.

We have provided these tables of accessory gene modules as a resource for scientists to explore and guide future research. For example, there are many missing gene annotations, particularly for accessory genes (69). Given that the most recent CAFA (Critical Assessment of Functional Annotation) study found that coexpression data were highly predictive of function, these modules could point to possible candidate functions that can be experimentally verified (70). In general, these supplemental tables (Tables S6 and S7) can provide a summary of the landscape for the accessory genes that have been researched and what remains to be explored. The expression statistics provided (mean and median expression values, intra- and intermodule distances, enrichment within KEGG pathways, and associated operons) could also be used to help scientists filter the set of genes to focus on.

## DISCUSSION

Through our analysis of core and accessory gene expression, we discovered that there is diverse transcriptional behavior within different gene groups between PAO1 and PA14 strains of *Pseudomonas aeruginosa*. Despite core genes being shared across strains, core genes differ in their transcriptional patterns, with some being highly stable and others being less so. The most stable core genes are less coexpressed with other PAO1 or PA14 accessory genes. One possibility is that PAO1 or PA14 accessory genes might change the transcriptional regulation of these core genes, which is why the most stable genes are less coexpressed with PAO1 and PA14 accessory genes (71). For example, it is known that insertion sequences, which often include accessory genes, can influence the expression of an existing gene via the disruption of the existing promoter or by the formation of a hybrid promoter (62–64). Additionally, a previous study found that accessory genes could modify the function of core genes between two strains of *Streptococcus pneumoniae* (71). Overall, by using the compendia of expression data, we were able to distinguish the types of core genes by their global transcriptional patterns, which will help inform future studies of regulatory mechanisms to better understand PAO1 and PA14 strain diversity.

In this analysis, we focused on examining core and accessory genes within *P. aeruginosa*; however, this analysis can be extended to other gene groups or organisms beyond *P. aeruginosa*. For example, bacteriophage genes, which are viral genes that are integrated into the bacterial genome, are another type of gene contained within the *P. aeruginosa* genome. Similar to other gene groups, changes in phage expression have been shown to affect phenotypes. For example, the upregulation of phage genes was found to promote biofilm development (72). Investigating phage gene expression within the transcriptional landscape would require generating a suitable reference genome that includes all phage genes, which does not currently exist and may be challenging to generate given the high rate of phage sequence evolution (73). This work could also be extended to other two-strain-type models for different microbes to help understand the phenotypic heterogeneity. Comparisons between two lineages of *Staphylococcus aureus* in Australia with different resistance phenotypes (74) and two *Plasmodium vivax* lineages with different geographical origins (75) may be well suited to this model. Other examples include comparing two groups of strains based on phenotypes: comparing rare versus common *Escherichia coli*

strain types (76, 77), comparing *Staphylococcus aureus* resistant versus sensitive strains (78), comparing *Acinetobacter baumannii* outbreaks in different hospital wards (79), or comparing symbiotic and nonsymbiotic species (80).

One limitation is that our analysis is limited to examining two strain types at a time. However, there are additional clades that we might want to consider. For example, for *P. aeruginosa*, there are many clinical strains in addition to PAO1 and PA14. To apply our approach to include clinical strains, we would need to generate a clinical reference genome to obtain a more accurate picture of the expression of clinical accessory genes. We would also need to determine the core and accessory annotations in order to account for an additional strain type. Finally, we would need to determine a new metric to assess stability since we will have more than two variables for our correlation calculation.

By leveraging expression compendia with hundreds to thousands of samples, this study reveals the complex relationship among and between PAO1 and PA14 core and accessory genes that should be considered when designing future regulatory analyses to further study *P. aeruginosa* strain-level diversity.

## MATERIALS AND METHODS

**PAO1 and PA14 sample compendia.** For these analyses, we used the PAO1-mapped and PA14-mapped RNA-seq compendia as described in the companion article by Doing et al. (52). Doing et al. provide filtered and median ratio (MR)-normalized compendia containing 2,333 samples mapped to 5,563 genes using cDNA sequences from the PAO1 reference genome (NCBI PAO1 cDNA accession number GCA_000006765.1) (81–83) and 5,891 genes using cDNA sequences from the PA14 reference genome (NCBI PA14 cDNA accession number GCA_000014625.1) (19). The filtering steps included the removal of sparse samples (i.e., samples having a high number of genes with zero counts) and those with median expression values of housekeeping genes that were too high or low (more details can be found in reference 52). MR normalization was then performed to enable comparisons across samples.

The SRA provided annotations for the strain type for ∼70% of the 2,333 samples. We used the expression activities of the accessory genes (i.e., genes that are specific to PAO1 or PA14) to validate these annotations and, for unannotated samples, to assess whether the sample was derived from strain PAO1, strain PA14, or a clinical isolate that differed from both strains. There were several advantages of using the expression activity instead of the metadata provided by the SRA. First, this approach allowed us to create a pipeline that can easily be extended to new data sets where metadata are not available. Despite the metadata being available in this case, the strain names require curation. Next, we were able to leverage more data as opposed to losing samples that have an unknown strain annotation (22% of the PAO1 predicted samples and 10% of the PA14 predicted samples have an unknown SRA annotation). A sample was included in the strain PAO1 compendium if the median gene expression value of PA14-only accessory genes was <25 counts and when that of the PAO1-only accessory genes was >25 counts. Similarly, a sample was included in the PA14 compendium if the median gene expression value of PA14-only accessory genes was >25 counts and when the median gene expression value of PAO1-only accessory genes was <25 counts. The threshold of 25 counts was used based on the distribution of PAO1-only accessory genes in SRA-annotated PAO1 samples compared with the distribution of SRA-annotated non-PAO1 samples. We defined the threshold to be one that separated PAO1 SRA-labeled from non-PAO1 SRA-labeled sample expression. Additionally, the use of a threshold of 25 counts allowed 99% of PAO1 and 98% of PA14 SRA-labeled samples to be assigned as PAO1 or PA14 based on expression. After applying these thresholds, we obtained a strain PAO1 sample compendium containing 129 experiments with 890 samples (median of 5 samples per experiment) and 5,563 genes. The strain PA14 sample compendium contained 59 experiments with 505 samples (median of 6 samples per experiment) and 5,892 genes. These two compendia included experiments that used a variety of different media and different mutant derivatives. The associated metadata, which included information such as medium, were curated using the GEOquery R package (84), using the same process as the one described in the companion article by Doing et al. (52), which was used to collect the metadata associated with the studies contained in the compendia. These metadata were manually curated to consolidate medium assignments into fewer categories. Additional manual curation was also applied to assign multiple medium annotations to a given experiment. For example, the data under BioProject accession number PRJNA169508 were annotated with LB medium and synthetic cystic fibrosis sputum medium (SCFM).

**PAO1 and PA14 array compendia.** We started with the *P. aeruginosa* gene array gene expression compendium defined previously by Tan et al. (47). This compendium contained 950 expression profiles that were measured upon the release of the GeneChip *P. aeruginosa* genome array (data freeze in 2014). We created two separate array compendia, equivalent to the PAO1 and PA14 sample compendia using RNA-seq data, by assigning samples to the PAO1 or PA14 array compendium based on their strain labels using the associated metadata. This resulted in a PAO1 array compendium with 436 samples and a PA14 array compendium with 99 samples.

**Core and accessory annotations.** For this study, we defined core genes as those present in both PAO1 and PA14, which are diverse isolates, acknowledging that this may include or exclude certain genes that would be differently categorized with random unbiased sequencing of strains, while accessory genes are those that are present in at least one strain. Here, we used the annotations of core genes obtained from the BACTOME website (53), where core genes are those that have at least 90% sequence

homology between the two strain types and where accessory genes are the remaining genes that are either PAO1 or PA14 specific. There were 5,361 core genes that were identified based on sequence homology between PAO1 and PA14 strains. Next, the accessory genes were identified by subtracting these 5,361 core genes from the PAO1 and PA14 reference genomes. We obtained 202 PAO1 (5,563 reference genes minus 5,361 core genes) and 530 PA14 (5,891 reference genes minus 5,361 core genes) accessory genes, respectively (see Table S8 in the supplemental material). The script for generating the core and accessory genes can be found at https://github.com/greenelab/core-accessory-interactome/blob/bf157513ab5041f8d0bf70766e79df2f06ea7bca/scripts/utils.py#L130.

**Distributions of PAO1 and PA14 sample compendia using principal-component analysis (PCA).** Samples from the PAO1 sample compendium (890 samples) and the PA14 sample compendium (505 samples) were selected from the original filtered and normalized RNA-seq compendium (2,333 samples) that was mapped to both the PAO1 and PA14 references. Our selection resulted in two data sets with 1,395 samples (PAO1 samples and PA14 samples combined) aligned to the PAO1 and PA14 references. We used the sklearn library to compress the selected PAO1 and PA14 data sets into the first 2 principal components for visualization (Fig. 1D and E; Fig. S1C and D). We then calculated the centroids of the PAO1 and PA14 sample compendia by taking the average of the first 200 principal components since 90% of the variance was explained by 200 components. Next, we measured the pairwise Euclidean distance between the two centroids to obtain the difference in the means between the two strain distributions. To calculate the spread of the PAO1 and PA14 sample compendia, we calculated the variance using the first 200 principal components.

**Correlation matrix.** When we generated Pearson correlation matrices for the PAO1 sample compendium and the PA14 sample compendium using the MR-normalized counts, we found that many gene pairs had a high correlation because many genes are related to the same pathway and therefore have similar expression profiles. This is consistent with the results of Myers et al. (85), who found that there can be an overrepresentation of genes associated with the same pathway (i.e., a large fraction of gene pairs represent ribosomal relationships). This very prominent signal makes it difficult to detect other signals. To remove this very dominant global signal in the data, we first $\log_{10}$ transformed the expression data and then applied a signal balancing technique called SPELL (86). The SPELL algorithm calculated the singular value decomposition (SVD) of the expression matrix to obtain the factorized set of matrices. We then applied the Pearson correlation to the SPELL-processed matrix, which is the factorized gene coefficient matrix $U$. This coefficient matrix represents how genes contribute to independent latent variables that capture the signal in the data where the variance of the variables is 1. The correlation of the SPELL matrix relates genes based on the gene coefficient matrix. In other words, the correlation of the SPELL matrix relates genes based on their contribution to singular vectors (SVs) that capture linear relationships between genes. A high correlation means that a pair of genes contributes similarly to a singular vector, which are the axes pointing in the direction of the spread of the data and capture how genes are related to each other. The advantage of using SPELL is that the gene contributions are more balanced so that redundant signals (i.e., many genes from the same pathway or genes that vary together) are represented by a few singular vectors as opposed to many samples. More balanced gene contributions also mean that more subtle signals can be amplified (i.e., genes related by a smaller pathway are also captured by a number of SVs similar to those for larger pathways). However, the one caveat is that SPELL can amplify noise; i.e., an SV that corresponds to some technical source of variability now has a weight similar to those of other real signals.

**Transcriptional stability of core genes.** For this analysis, we started with a PAO1 core gene, say PA0001. First, we selected the correlation scores for all genes related to PA0001. Next, we selected the homologous PA14 gene, PA14_00010, and likewise pulled the correlation scores for how genes were related to PA14_00010. Next, we calculated the Pearson correlation between the two correlation profiles to obtain a transcriptional stability score. We repeated this analysis for all core genes. We removed 5 PAO1 core genes and 10 PA14 core genes that had ambiguous homolog mapping across strains (i.e., a single gene identifier from one strain mapped to multiple gene identifiers from the other strain). High scores (genes in the top 5%) indicated that the transcriptional relationships of the core genes were consistent or the most stable, whereas low scores (genes in the bottom 5%) indicated that the core genes were the least stable. This analysis was performed using both the PAO1 sample and PA14 sample RNA-seq compendia (this study) as well as the array compendia (436 PAO1 samples and 99 PA14 samples) derived from the array compendium described previously by Tan et al. (47). The processing steps described above were applied to both compendia.

**Core gene stability versus location consistency.** For each of the most stable core genes, we selected the neighboring core genes (10 genes upstream of the most stable core gene and 10 genes downstream) in PAO1 and PA14. We then calculated the overlap between the 20 neighboring core genes in PAO1 and PA14 and asked, How many of the 20 neighboring core genes in PAO1 are homologous to the neighboring genes in PA14? If there was a large overlap between the neighboring genes across strains, this would indicate that the most stable core gene was located in the same genomic region in PAO1 and PA14. We repeated this calculation for all of the most stable core genes. We also performed this calculation starting with the least stable core genes. We performed this analysis using 5,349 core genes using the PAO1 and PA14 sample compendia. We also performed this analysis using 5,339 core genes using the PAO1 and PA14 array compendia.

**KEGG enrichment analysis.** The goal of an enrichment analysis is to detect coordinated changes in prespecified sets of related genes (i.e., those genes that are in the same pathway or share the same Gene Ontology [GO] term). Here, we used one-sided Fisher's exact test. This test asked if there was a significant overrepresentation of KEGG pathways in the most and least stable core gene sets. Specifically, this method tested if the proportion of KEGG annotations within the most stable core gene set was higher than expected

compared to the proportion of KEGG genes in the total data set. The pathways used in this analysis were downloaded from the KEGG website (http://www.genome.ad.jp/kegg/) using the Python Bio.KEGG library. These pathways can be found in the associated repositories at https://github.com/greenelab/core-accessory-interactome/blob/master/3_core_core_analysis/pao1_kegg_annot.tsv and https://github.com/greenelab/core-accessory-interactome/blob/master/3_core_core_analysis/pa14_kegg_annot.tsv.

**Relationships between core and accessory genes.** Here, we examined the genes that the most and least stable core genes are related to. For this analysis, we started with the most stable core genes and asked, Is the most highly correlated gene core or accessory? For a given stable core gene, we obtained a list of genes sorted by their correlation score, making sure to remove any genes that were cooperonic with the start gene. The operon data were provided by Geoff Winsor and include computationally predicted annotations from DOOR (87) as well as curated annotations from PseudoCAP (82, 83). DOOR uses sequence features, including intergenic distance, conservation of neighboring genes across genomes, and phylogenetic distance, to classify genes as being cooperonic (87). Their method was trained on experimentally validated operons. In addition to DOOR annotations, PseudoCAP provides annotations based on manually reviewed literature (83). We repeated this selection and filtering for all of the most stable core genes so that for all of the most stable core genes, we have a list of the top 10 most coexpressed genes. We can then calculate the proportion of the most coexpressed genes that are accessory. This proportion is normalized by the proportion of accessory genes within the whole genome. The resulting score is near 1.0 if the proportion of accessory gene relationships is no different from the baseline proportion of accessory genes in the genome. If the resulting score is higher or lower than 1.0, then the proportion of accessory gene relationships is higher or lower than expected compared to the baseline. We repeated this calculation for the second most coexpressed genes, the third most coexpressed genes, and so on, so that we have a fold change over a random value for each top 10 coexpression ranking. We also performed the same calculation for the 11th most coexpressed genes and beyond to act as a baseline. This analysis was repeated starting with the least stable core genes as well.

**Accessory-accessory module detection.** To obtain accessory-accessory modules, we started with the gene expression of only accessory genes, which included 202 PAO1-specific genes for the PAO1 sample compendium and 530 PA14-specific genes for the PA14 sample compendium. To define a set of modules, we first applied Pearson correlations to the SPELL-processed expression matrices (see above) to obtain correlation scores for how similar each pair of genes is based on the transcriptional profiles. Next, we applied clustering to the correlation matrices. The clustering method used is called affinity propagation (68). Affinity propagation is a graph-based clustering algorithm similar to k-means but without the need to specify the number of clusters *a priori*. Affinity propagation finds "exemplars," which are members of the input (or genes in our case) that are representative of clusters. These exemplars are equivalent to "centroids" in k-means. Exemplars are determined by an exchange of messages, which consists of "responsibility" and "availability." Responsibility quantifies how well suited gene X is to being an exemplar based on how similar other genes are to gene X and how available gene X is. Availability measures how likely a gene is to choose gene X as its exemplar. Availability is calculated as the responsibility of gene X toward itself and the responsibility of gene X toward all other genes. The clustering returned 28 modules for PAO1 and 70 modules for PA14. We validated that these accessory modules tended to correlate with known operons based on manual inspection. While we did not perform a thorough evaluation to determine that this clustering approach is the best one, we found that the modules appear to be biologically relevant and can serve as a stepping stone for future research.

**Software.** All scripts used in these analyses are available in the GitHub repository (https://github.com/greenelab/core-accessory-interactome) under an open-source license to facilitate the reproducibility of these findings. The repository's structure is described in the Readme file. The virtual environment was managed using conda (version 4.9.2), and the required libraries and packages are defined in the environment.yml file.

**Data availability.** The expression data can be found in the repository at https://osf.io/vz42h/. This repository includes the PAO1-mapped and PA14-mapped RNA-seq compendia (2,333 samples) described in the companion article by Doing et al. (52). This repository contains the following strain-specific compendia: the PAO1 sample compendium (890 samples) and the PA14 sample compendium (505 samples). Additionally, the median gene expression values of PAO1-only and PA14-only accessory genes can be found at https://osf.io/vdp9u.

## SUPPLEMENTAL MATERIAL

Supplemental material is available online only.

**FIG S1**, EPS file, 1.6 MB.
**FIG S2**, EPS file, 1.7 MB.
**TABLE S1**, XLSX file, 1.4 MB.
**TABLE S2**, XLSX file, 0.02 MB.
**TABLE S3**, XLSX file, 0.02 MB.
**TABLE S4**, XLSX file, 0.4 MB.
**TABLE S5**, XLSX file, 0.6 MB.
**TABLE S6**, XLSX file, 0.1 MB.
**TABLE S7**, XLSX file, 0.1 MB.
**TABLE S8**, XLSX file, 0.02 MB.

## ACKNOWLEDGMENTS

We thank Geoff Winsor for providing operon data that we used for our analysis relating core genes to other accessory genes. We also thank Jake Crawford and Natalie Davidson for reviewing the software associated with this work and providing valuable feedback.

This work was supported by grants from the Gordon and Betty Moore Foundation (GBMF4552 to C.S.G.), the Cystic Fibrosis Foundation (HOGAN19GO to D.A.H., GREENE21GO to C.S.G., and STANTO19R0 to S.L.N.), and the Flatley Foundation. Finally, this work was supported by the National Institutes of Health (NIH) through awards NIDDK P30-DK117469, P30 DK117469, and R01 HL151385.

A.J.L., Formal Analysis, Investigation, Methodology, Project Administration, Software, Visualization, Writing – Original, and Writing – Review. G.D., Data Curation and Writing – Review. S.L.N., Data Curation and Writing – Review. T.R., Writing – Review. D.A.H., Conceptualization, Funding Acquisition, Methodology, Supervision, Writing – Original, and Writing – Review. C.S.G., Conceptualization, Funding Acquisition, Methodology, Supervision, and Writing – Review.

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
