## [Reviewer comments · mSystems]

Compendium-wide analysis of *P. aeruginosa* core and accessory genes reveal transcriptional patterns across strains PAO1 and PA14

Alexandra Lee, Georgia Doing, Samuel Neff, Taylor Reiter, Deborah Hogan, and Casey Greene

Corresponding Author(s): Casey Greene, University of Colorado Anschutz Medical Campus

Review Timeline:

Submission Date:	April 12, 2022
Editorial Decision:	June 14, 2022
Revision Received:	August 16, 2022
Editorial Decision:	October 5, 2022
Revision Received:	October 11, 2022
Accepted:	October 12, 2022

Editor: Elizabeth Shank

Reviewer(s): The reviewers have opted to remain anonymous.

Transaction Report:

DOI: <https://doi.org/10.1128/msystems.00342-22>

June 14, 2022

Prof. Casey S Greene
University of Colorado Anschutz Medical Campus
Biochemistry
Aurora, CO

Re: mSystems00342-22 (Compendium-wide analysis of *P. aeruginosa* core and accessory genes reveal more nuanced transcriptional patterns)

Dear Prof. Casey S Greene:

Thank you for submitting your manuscript to mSystems. We have now completed our review and the reviewers have indicated that the work is interesting and publishable in principle in mSystems. However, as you will see, the reviewers make some important suggestions with regard to some of the analyses, descriptions, and language of the manuscript. As you know, I have also now spoken with Deb Hogan with respect to this manuscript and its companion manuscript, where we discussed the possible changes to be made to ensure both manuscripts stand alone and that their unique contributions are clear.

Thank you for the privilege of reviewing your work. Below you will find the comments from the reviewers as well as instructions from the mSystems editorial office regarding resubmission.

Preparing Revision Guidelines

Sincerely,

Elizabeth Shank

Editor, mSystems

Journals Department
American Society for Microbiology
1752 N St., NW

Reviewer comments:

Reviewer #1 (Comments for the Author):

The manuscript by Lee and colleagues uses a new gene expression compendium from *Pseudomonas aeruginosa* to compare the expression of core and accessory genes in the two main lab workhorse strains PAO1 and PA14. Through this comparison they identified subsets of core genes with stable expression patterns across strain types as well as those with less stable expression patterns. They also found that core genes with the least stable expression patterns were more likely to be co-expressed with accessory genes than core genes with stable expression patterns. They highlight additional interesting findings amongst their data, including observations about strains containing the type III secretion system effectors ExoS or ExoU. In particular, they find that for each effector gene there are both core-genes that are co-expressed with them that are common to both or unique to each. In general, the study provides new information regarding the conservation of gene expression signatures amongst *P. aeruginosa* isolates grown under a variety of conditions. This information should prove a useful resource to the *Pseudomonas* community; the data could be used for a variety of purposes, such as improving the accuracy of operon prediction or providing possible transcriptional differences to explain observed phenotypic differences between strain types.

Minor points

1. Line 398-400. In relation to the PAO1- and PA14-mapped RNA-seq compendia. Samples based on too high or low median expression of housekeeping genes were discarded. What's the rationale for discarding these as they could be quantifying biologically relevant phenomena?
2. Line 112. State what abbreviation SRA stands for.
3. Line 181-182 "stable core genes also included genes within the type VI secretion system (T6SS) and genes in the Hcp Secretion island-I (H1-T6SS)." I don't quite get the distinction between the two. Doesn't the Hcp secretion island encode a type VI secretion system?
4. Line 183. "TVI genes". Are these T6SS genes?
5. Were the two copies of the pyocanin gene operons (phz genes) taken into account during these studies?
6. Section beginning line 268. For the discussion in differences in the co-expression patterns of core genes with ExoS and ExoU, it might be worth highlighting that these effector genes are expressed from different genomic locations in the respective strains.

Reviewer #2 (Comments for the Author):

The manuscript titled "Compendium-wide analysis of *P. aeruginosa* core and accessory genes reveal more nuanced transcriptional patterns" with control number mSystems00342 describes the development of a compendium of expression data from the most common *Pseudomonas aeruginosa* isolates with expression data, PAO1 and PA14. The manuscript is much improved in the description of the methodology, however as outlined below there are a number of critical steps in the analysis pipeline that are not described or lack detail enough to evaluate. A number of critical issues for the authors to consider include the viability of the mapping algorithm, the evaluation of the operon studies (defining what and how the operons are identified versus co-expressed and co-located - these are not synonyms of each other, yet seem to be approached in the same way), as well as the mapping to only two common, but not representative isolates of *P. aeruginosa*. Additionally, the data included companion paper should be part of this study, as the definition of only the core or only the accessory genome in either case is not complete or comprehensive.

Major Comments:

The number of samples examined in this manuscript and the companion submission is not the same. Please indicate why this is the case, what different criteria were used in the selection of the datasets to include and how this impacts to outcomes of each study. Additionally, the accessory genome analysis should be included in this study. While I appreciate that there is a lot of work in each, the datasets are truly meant to be in the same study and separating them diminishes the impact of each to the point of not being able to stand alone.

The authors point out that Salmon is not a good aligner for prokaryotic transcriptomes as it does not handle the biology of the system well (the authors mention operons and short transcripts). It is unclear if the CLC workbench has the same inherent

biases that prokaryotic transcriptomes are "less complex" than eukaryotic transcriptomes and thus the tools will work fine. There are no transcriptional aligners that deal with the intricacies of prokaryotic transcriptomes. By changing the flag to "unpaired" the authors are not representing the publicly available data correctly and they have no way to know which is truth.

The description of the methodology remains - for example the authors suggest that there was mapping to "PAO1 and PA14 cDNA reference genomes and validating these compendia using expression patterns gene sets comprised of core genes present in both strains" (lines 246-247), however it is not clear how these cDNA reference genomes are created and how the core genes were identified and selected. Additionally, the number of samples changes throughout the manuscript without explanation as to how and why these were sub-selected (2852 on line 114 , 2333 on line 245, 2332 on line 597 etc). There is a suggestion on line 340/341 that suggest samples were filtered, but it is not clear why or on what criteria? This is critical information to the evaluation of this submission.

As previously suggested the authors want to title, introduction and discussion to demonstrate that the data is widely applicable to all *P. aeruginosa*, but the data is limited to PAO1 and PA14 and thus does not really support this broad characterization. For example, what happens if the reference genomes are changed? If this method is truly agnostic of the reference genome the core should be similar and thus the results should be similar. The creation of a *P. aeruginosa* pan genome for mapping would provide an acceptable pseudo-genome for mapping and address the variability of *P. aeruginosa*.

Descriptions of genes in operons - it is unclear how the authors are defining operons and what that definition is based on - is it a reference, database or just co-localized and co-expressed (which is not the definition of an operon).

In Figure 4C/D the maintained and removed samples provide troubling insight into the effectiveness of the selection criteria as there does not seem to be a clear delineation of the samples that should be included or excluded. Another interpretation of this data is that the PCA plots are not an effective way to display this data. It is unclear why this data is mixed together.

Minor Comments:

PCA plots need to have the amount of variation that is accounted for by each of the axis included?

Lines 294-297 - "Taken together, we found that Salmon mapped data produced DE results similar to those derived from published count tables obtained using other methods across data produced by our lab and others using different strains" this statement needs some type of validation - supplemental dataset or something addition to show how similar the analyses are? Are the number of genes increased/decreased in expression relative to a control the same? Or are the gene levels similar? Since there is no control for this expression how is this normalized?

October 5, 2022

Prof. Casey S Greene
University of Colorado Anschutz Medical Campus
Biochemistry
Aurora, CO

Re: mSystems00342-22R1 (Compendium-wide analysis of *P. aeruginosa* core and accessory genes reveal nuanced transcriptional patterns)

Dear Prof. Casey S Greene:

Thank you for submitting your manuscript to mSystems. You will note that although the reviewers have a difference of opinion concerning the independence of your manuscript from its companion submission, we have completed our review and I am pleased to inform you that, in principle, we expect to accept it for publication in mSystems. However, acceptance will not be final until you have adequately addressed the remaining minor correction. Specifically, ensure that you have an explicitly labeled Data Availability paragraph at the end of your manuscript to comply with ASM's Data Policy (<https://journals.asm.org/open-data-policy>). The necessary information largely appears embedded in the current manuscript methods section but is not appropriately labeled as "Data Availability".

Preparing Revision Guidelines

Sincerely,

Elizabeth Shank

Editor, mSystems

Journals Department
Reviewer comments:

Reviewer #2 (Comments for the Author):

All of my prior comments were minor and in their revised manuscript the authors have fully addressed them.

Reviewer #3 (Comments for the Author):

The revised manuscript by Lee et al with title "Compendium-wide analysis of *P. aeruginosa* core and accessory genes reveal nuanced transcriptional patterns" and control number mSystems00342-22R1 is improved; however there are a number of issues that the authors did not address in the revision as pointed out in the previous revision or the revised version has brought to light other inconsistencies in either the data or the analysis.

I will provide this comment as I have provided it before. While there are differences between these manuscripts, neither can truly stand alone, which suggests that the studies are not truly independent and should be within a single manuscript. The inclusion of the core genome in one paper and the accessory genome (as defined by the authors) in the second manuscript really diminishes both.

As with the companion submission mSystems00341-22R1 the interpretation of the data is fundamentally limited to the two genomes used. For example, when the authors describe "all strains" on line 1517, then clarify on line 1518 that all =2. The compendium wide analysis as indicated in the title is really limited to two isolates. While diverse isolates, are not representative of the wide diversity of *P. aeruginosa*. This results in an overstatement of the findings and limited utility of the methodology and data as currently presented. The authors cite the array compendium that they have previously published; however the RNAseq technology is different and thus the methods used for that analysis are not valid for this technology.

As previously indicated the definition of "nuanced" in the title is vague and never defined. How and why is it nuanced?

Reviewer comments:

Reviewer #2 (Comments for the Author):

All of my prior comments were minor and in their revised manuscript the authors have fully addressed them.

- Great. Thank you for all your feedback.

Reviewer #3 (Comments for the Author):

The revised manuscript by Lee et al with title "Compendium-wide analysis of *P. aeruginosa* core and accessory genes reveal nuanced transcriptional patterns" and control number mSystems00342-22R1 is improved; however there are a number of issues that the authors did not address in the revision as pointed out in the previous revision or the revised version has brought to light other inconsistencies in either the data or the analysis.

I will provide this comment as I have provided it before. While there are differences between these manuscripts, neither can truly stand alone, which suggests that the studies are not truly independent and should be within a single manuscript. The inclusion of the core genome in one paper and the accessory genome (as defined by the authors) in the second manuscript really diminishes both.

As with the companion submission mSystems00341-22R1 the interpretation of the data is fundamentally limited to the two genomes used. For example, when the authors describe "all strains" on line 1517, then clarify on line 1518 that all =2. The compendium wide analysis as indicated in the title is really limited to two isolates. While diverse isolates, are not representative of the wide diversity of *P. aeruginosa*. This results in an overstatement of the findings and limited utility of the methodology and data as currently presented. The authors cite the array compendium that they have previously published; however the RNAseq technology is different and thus the methods used for that analysis are not valid for this technology.

- We have revised the text to clarify the specific definition of core genes that we're using in our study. The new text now reads: "*For this study, we defined core genes as those present in both PAO1 and PA14, which are diverse isolates, acknowledging that this may include or exclude certain genes that would be differently categorized in a random, unbiased sequencing of strains.*"

As previously indicated the definition of "nuanced" in the title is vague and never defined. How and why is it nuanced?

- We agree that this is a vague description. We have updated the title to "Compendium-wide analysis of *P. aeruginosa* core and accessory genes reveal transcriptional patterns across strains PAO1 and PA14"

October 12, 2022

Prof. Casey S Greene
University of Colorado Anschutz Medical Campus
Biochemistry
Aurora, CO

Re: mSystems00342-22R2 (Compendium-wide analysis of *P. aeruginosa* core and accessory genes reveal transcriptional patterns across strains PAO1 and PA14)

Dear Prof. Casey S Greene:

Your manuscript has been accepted, and I am forwarding it to the ASM Journals Department for publication. For your reference, ASM Journals' address is given below. Before it can be scheduled for publication, your manuscript will be checked by the mSystems production staff to make sure that all elements meet the technical requirements for publication. They will contact you if anything needs to be revised before copyediting and production can begin. Otherwise, you will be notified when your proofs are ready to be viewed.

Publication Fees:

If you would like to submit a potential Featured Image, please email a file and a short legend to mSystems@asmusa.org. Please note that we can only consider images that (i) the authors created or own and (ii) have not been previously published. By submitting, you agree that the image can be used under the same terms as the published article. File requirements: square dimensions (4" x 4"), 300 dpi resolution, RGB colorspace, TIF file format.

We recognize that the video files can become quite large, and so to avoid quality loss ASM suggests sending the video file via <https://www.wetransfer.com/>. When you have a final version of the video and the still ready to share, please send it to mSystems staff at mSystems@asmusa.org.

Sincerely,

Elizabeth Shank
Editor, mSystems

Journals Department
Table S8: Accept
Table S1: Accept
Figure S1: Accept
Figure S2: Accept
Table S3: Accept
Table S2: Accept
Table S4: Accept
Table S5: Accept
Table S6: Accept
Table S7: Accept